# Shape of my heart: Cardiac models through learned signed distance functions

**Jan Verhülsdonk**[*1]       VERHUELSDONK@IAM.UNI-BONN.DE

**Thomas Grandits**[*2,4]       THOMAS.GRANDITS@UNI-GRAZ.AT

**Francisco Sahli Costabal**[3]       FSC@ING.PUC.CL

**Thomas Pinetz**[1]       PINETZ@IAM.UNI-BONN.DE

**Rolf Krause**[4,5]       ROLF.KRAUSE@USI.CH

**Angelo Auricchio**[4,6]       ANGELO.AURICCHIO@EOC.CH

**Gundolf Haase**[2]       GUNDOLF.HAASE@UNI-GRAZ.AT

**Simone Pezzuto**[4,7]       SIMONE.PEZZUTO@UNITN.IT

**Alexander Effland**[1]       EFFLAND@IAM.UNI-BONN.DE

[1] *Institute for Applied Mathematics, University of Bonn, Germany*

[2] *Department of Mathematics and Scientific Computing, University of Graz, Austria*

[3] *Institute for Biological and Medical Engineering, Pontificia Universidad Católica de Chile, Chile*

[4] *Center for Computational Medicine in Cardiology, Università della Svizzera italiana, Switzerland*

[5] *FernUni Schweiz, Brig, Switzerland*

[6] *Instituto Cardiocentro Ticino, EOC, Switzerland*

[7] *Department of Mathematics, University of Trento, Italy*

**Editors:** Accepted for publication at MIDL 2024

## Abstract

The efficient construction of anatomical models is one of the major challenges of patient-specific in-silico models of the human heart. Current methods frequently rely on linear statistical models, allowing no advanced topological changes, or requiring medical image segmentation followed by a meshing pipeline, which strongly depends on image resolution, quality, and modality. These approaches are therefore limited in their transferability to other imaging domains. In this work, the cardiac shape is reconstructed by means of three-dimensional deep signed distance functions with Lipschitz regularity. For this purpose, the shapes of cardiac MRI reconstructions are learned to model the spatial relation of multiple chambers. We demonstrate that this approach is also capable of reconstructing anatomical models from partial data, such as point clouds from a single ventricle, or modalities different from the trained MRI, such as the electroanatomical mapping (EAM).

**Keywords:** Deep Signed Distance Function, Shape Reconstruction, Cardiac Modeling, Lipschitz Regularized Network

## 1. Introduction

Modern personalized precision medicine frequently targets patient-specific therapies with improved therapy outcomes, reduced intervention times, and thus lower costs. In the case of cardiac personalized treatment, this necessitates complex simulation models from patient

---

[*] Contributed equally

data Corral-Acero et al. (2020). This future vision of personalized cardiac treatment relies on generated 3D models, which should represent the anatomy of the corresponding patient.

Current methods to generate anatomical models usually require computed tomography (CT) or magnetic resonance (MR) images, which are segmented and subsequently meshed (Strocchi et al., 2020a). Yet, while automatic cardiac image segmentation is well-researched and constantly improving thanks to machine learning (Painchaud et al., 2020; Campello et al., 2021; Zhuang et al., 2022), it is limited to a specific image modality and resolution. Other modalities such as electroanatomical catheter mapping (Bhakta and Miller, 2008) are difficult to fit within standard frameworks, albeit they are very important for patient-specific modelling (Ruiz-Herrera et al., 2022).

In this work, we propose the use of implicitly learned representations through signed distance functions (SDFs). We start from the *DeepSDF* method (Park et al., 2019), which encodes SDF-based surfaces through a decoder-only neural network with a small-dimensional input latent code. The resulting neural network is a cardiac shape atlas. Shape inference can be achieved from location measurements such as electro-anatomical mappings and is applicable to arbitrary resolutions. Additionally, interpolation in the latent space yields shape interpolation in the physical space, a useful feat to create novel shapes. Differently from the original *DeepSDF* method, here we take advantage of Lipschitz-regularized neural networks (Liu et al., 2022) so to avoid overfitting of the network and to enforce smooth interpolations between samples in the latent space. This method is especially efficient for learning on relatively small training sets as considered here.

We demonstrate that this Lipschitz-regularized *DeepSDF* architecture is suited for constructing cardiac models by learning from a database of 44 publicly available, post-processed cardiac models. The resulting *DeepSDF* model is designed to encode bi-ventricular shapes and also infer new shapes from sparse point clouds, even if only partial domains are available (e.g., only endocardial measurements of a single chamber often encountered in electroanatomical mappings). The proposed method shares some similarities with the very recent work by Sander et al. (2023); however, our approach exhibits several major advantages compared with this paper: 1. applying a *Lipschitz-regularized* version of *DeepSDF* methodology to the cardiac domain on the example of bi-ventricular models and demonstrating advantageous accuracy in the presence of sparse data in comparison to state-of-the-art methods, 2. thanks to the multi-chamber approach both topological constraints are implicitly incorporated and adjacent anatomical structures benefit from additional available data, which might also aid uncertainty quantification in future research, 3. our approach is more robust against measurement errors modeling noise due to the iterative noise estimation and the additional prior penalization, which allows for the reconstruction of cardiac shapes from in-vivo measurements of an electroanatomical mapping procedure.

## 2. Related work

Surface reconstruction from point clouds is an intensively researched topic and provides a variety of possible approaches (Ma et al., 2022). Classical approaches assume no prior knowledge of the given point clouds, but in turn, require dense point clouds and are often sensitive to noise (Carr et al., 2001; Kazhdan et al., 2006; Kazhdan and Hoppe, 2013; Ummenhofer and Brox, 2015; Fuhrmann and Goesele, 2014). Modern deep-learning approaches

try to overcome these limitations by learning prior information about shapes using learned SDFs (Park et al., 2019; Hanocka et al., 2020), occupancy fields (Mescheder et al., 2019), level sets (Michalkiewicz et al., 2019) or implicit fields (Chen and Zhang, 2019).

In this context, the *DeepSDF* method is one of the most active fields of research. Such methods usually try to infer a multitude of shapes from at least partially dense point clouds. However, this method is prone to overfitting the data, thus often requiring regularization both in the latent space, as well as in the neural network by imposing dropout rules (Park et al., 2019). In Liu et al. (2022), Lipschitz-regularized linear layers are proposed, which are able to overcome some of these limitations. Alternative forms of regularization are, for instance, based on the eikonal equation (Gropp et al., 2020), other methods rely on local SDFs (Chabra et al., 2020; Jiang et al., 2020; Tretschk et al., 2020; Erler et al., 2020), neural-pulls (Baorui et al., 2021), or learned unsigned distance functions (Chibane et al., 2020). For further methods and comparisons, we refer the reader to Ma et al. (2022).

*Statistical Shape Models* (SSMs) are classical methods for (cardiac) shape reconstruction, which are tailored to the whole heart (Ecabert et al., 2008; Hoogendoorn et al., 2013; Lötjönen et al., 2004; Ordas et al., 2007; Unberath et al., 2015; Zhuang et al., 2010), the atria (Nagel et al., 2021), or the ventricles (Bai et al., 2015; Petersen et al., 2017). Only a few works have shown that deep learning methods can be applied to the cardiac domain to reconstruct cardiac shapes from point clouds (Beetz et al., 2023; Kong et al., 2023; Xiong et al., 2022; Sander et al., 2023; Alblas et al., 2023; Wang et al., 2021), whereas most of the learning based methods heavily rely on image input (Beetz et al., 2022; Kong et al., 2021).

## 3. Method

In this section, we comment on the representation of cardiac shapes by learned SDFs and provide details about training and shape completion.

### 3.1. General setting

We assume that each shape describing the heart is volumetrically represented as an SDF given by $f_S : \mathbb{R}^3 \to \mathbb{R}$ mapping from spatial points $\mathbf{x} \in \mathbb{R}^3$ to their respective signed distances $s \in \mathbb{R}$. These signed distances encode the signed projection distance at a point $\mathbf{x}$ to the surface (negative values inside, positive values outside). In such a way, the cardiac surface is represented as the zero-level set of this SDF, i.e., $\{\mathbf{x} \in \mathbb{R}^3 \colon f_S(\mathbf{x}) = 0\}$. A straightforward approach to learning such an SDF for arbitrary shapes involves representing the SDF by a neural network $S_\theta : \mathbb{R}^3 \to \mathbb{R}$, in which case the SDF is fully defined by the network's architecture and weights $\theta$. However, modeling surfaces solely through their neural network weights would necessitate a separate SDF and subsequent neural network for each individual surface, while also neglecting any similarities between the available training shapes. Following (Park et al., 2019), we instead model all shapes using a single neural network and additionally provide this network with a $d$-dimensional latent representation $\mathbf{z} \in \mathbb{R}^d$ of the shape as an additional input, together with the aforementioned spatial coordinates, i.e. $S_\theta : \mathbb{R}^{3+d} \to \mathbb{R}$. Hence, we encode multiple heart geometries with the same network, but different latent codes. We propose to model each of the left and right endocardial and epicardial surfaces with a shared latent code and neural network by letting the DeepSDF simultaneously estimate a signed distance for each of the closed bi-ventricular surfaces $\mathbf{s} =$

$(s_1, s_2, s_3, s_4)^\top \in \mathbb{R}^4$. In summary, the learned DeepSDF is given by the vector-valued function $\mathbf{S}_\theta : \mathbb{R}^{3+d} \to \mathbb{R}^4$.

## 3.2. Training and network architecture

The anatomical samples $X_i := (\mathbf{x}_k, \mathbf{s}_k)_{k=1}^{K_i}$ used for learning are $K_i$ pairs of spatial coordinates $\mathbf{x}_k$ with their sampled signed distance vector $\mathbf{s}_k$ for the $i$-th biventricular shape. The set of all $N$ available anatomical bi-ventricular samples is denoted as $X := (X_i)_{i=1}^N$. To each anatomical shape $i$, we associate a coupled latent code representation $\mathbf{z}_i \in \mathbb{R}^d$ also learned from data. We denote the set of learned latent codes as $Z := (\mathbf{z}_i)_{i=1}^N$. Our goal during training is to minimize the mismatch between the sampled signed distances and the ones estimated by the network through a simple quadratic loss term. Additionally, to restrict the latent codes $Z$, we follow (Park et al., 2019) and assume a zero mean Gaussian prior distribution with covariance $\sigma^2 I$ on the latent codes, which gives rise to the loss term

$$\mathcal{L}(\theta, Z) = \frac{1}{N} \sum_{i=1}^N \sum_{(\mathbf{x}_k, \mathbf{s}_k) \in X_i} \frac{1}{4K_i} \|\mathbf{S}_\theta(\mathbf{x}_k, \mathbf{z}_i) - \mathbf{s}_k\|^2 + \frac{1}{\sigma^2} \|\mathbf{z}_i\|_2^2.$$

The parameter $\sigma$ is used to balance between reconstruction accuracy for the training shapes and regularity in the latent space, which is essential for shape completion as described in Section 3.3. See Section 4.1 for the choice of $\sigma$ in our model. In Liu et al. (2022), it was shown that learning $\mathcal{L}$ directly may result in overfitting and might provide poor interpolation properties for the latent space. To overcome these issues, a Lipschitz penalization on the network was proposed in order to better control the smoothness. For this purpose, the Lipschitz bound $L$ of the network is estimated by $L = \prod_{i=1}^M \|W_i\|_p$ for an $M$-layer deep network, where $W_i$ are the network weights of the $i$-th layer. We closely follow the implementation of (Liu et al., 2022) where the linear layers of our network are replaced with Lipschitz-normalized layers. For each layer an additional weight $c_i$ is introduced such that softplus$(c_i) = \ln(1 + e^{c_i})$ serves as an upper bound softplus$(c_i) \geq \|W_i\|_p$ for the Lipschitz constant. Integrating this Lipschitz-regularization into our previous loss-functional $\mathcal{L}$ leads to our finally used cost-functional

$$J(\theta, Z) = \mathcal{L}(\theta, Z) + \alpha \prod_{c_i \in C(\theta)} \text{softplus}(c_i), \tag{1}$$

where $C(\theta) = (c_i)_{i=1}^M$ denotes the network parameter dependent per-layer Lipschitz bounds.

As mentioned, for samples $X_i := (\mathbf{x}_k, \mathbf{s}_k)_{k=1}^{K_i}$ the network inputs are the latent code $\mathbf{z}_i$ and the spatial coordinates $\mathbf{x}_i$ of the sample point. Following Liu et al. (2022) the spatial coordinates are multiplied by the factor $C_s = 100$ to balance the Lipschitz regularity of the latent space and the spatial coordinates. We use 5 hidden layers with 256 neurons and tanh activation functions. In the third hidden layer, the latent code and the spatial coordinate are concatenated to the output of the previous layer. The last layer is a linear layer with four signed distance functions as output, one for each of the left, right, epi- and endocardial surfaces. The latent code size was chosen as $d = 64$. A schematic representation of the network is shown in Figure 1 (left).

For the training of our network, we built a public shape library of watertight 3D shapes for endo-/epicardium of the left/right ventricles (Grandits et al., 2024) (4 shapes in total)

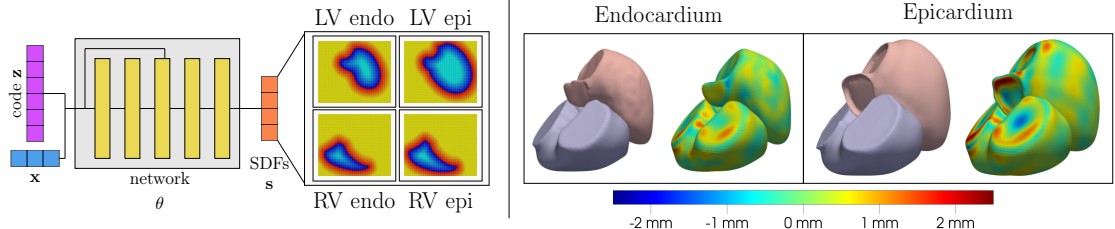

**Figure 1:** Left: a schematic representation of the employed DeepSDF. Right: comparison of ground truth meshes (LV in red, RV in blue) with reconstruction on the training dataset. The reconstructed meshes are color-coded with the signed distance to the ground truth mesh in mm.

based on Rodero et al. (2021); Strocchi et al. (2020b). Further details on the construction of the training data is provided in appendix A.

### 3.3. Shape completion

After learning the SDF network, a new anatomical bi-ventricular shape can be inferred from sparse point clouds of any combination of surfaces or given signed distance. For this, we consider $K$ given samples consisting of triplets $\tilde{Y}$ of spatial coordinates $\mathbf{x}_k \in \mathbb{R}^3$, a single signed distance $s_k \in \mathbb{R}$, and an index of the surface $j_k \in \{1, 2, 3, 4\}$. If a point $\mathbf{x}_k$ lies on the surface $j_k$, then $s_k = 0$. Finding the bi-ventricular reconstruction of a point cloud thus reduces to finding its latent code representation $\mathbf{z}$ by minimizing the following problem

$$\min_{\mathbf{z}} \frac{1}{K} \sum_{(\mathbf{x}_k, s_k, j_k) \in \tilde{Y}} \left( (\mathbf{S}_\theta(\mathbf{x}_k, \mathbf{z}))_{j_k} - s_k \right)^2 + \frac{\beta}{\sigma^2} \|\mathbf{z}\|_2^2, \tag{2}$$

where the subscript refers to the vector component and $\beta$ is an additional weight that is increased depending on the noise of the input point cloud. For noise-free point clouds, we set $\beta = 1$ to obtain the maximum-a-posterior (MAP) estimation (Park et al., 2019).

For noisy input point clouds, the expected data loss of a perfect reconstruction can be approximated with the variance of the noise $\xi$. We set $\beta = C_b \xi$, where $C_b = 100$ is a chosen scaling factor. In the case of real-world data, the underlying noise structure is unknown and estimated as follows: We start with a reconstruction $\mathbf{S}_\theta^0$ for an arbitrary initial noise estimation $\bar{\xi}_0$. We then iteratively estimate the *empirical variance* $\bar{\xi}_{n+1}^2$ based on $\mathbf{S}_\theta^n$ via

$$\bar{\xi}_{n+1}^2 = \frac{1}{K-1} \sum_{(\mathbf{x}_k, s_k, j_k) \in \tilde{Y}} \left( (\mathbf{S}_\theta^n(\mathbf{x}_k, \mathbf{z}))_{j_k} - s_k \right)^2. \tag{3}$$

This estimation exhibits a high experimental convergence rate (see appendix Section F).

### 4. Experiments

In this section, we state details of the training procedure and the network architecture. Moreover, we introduce distinct metrics for the evaluation of the surface reconstruction quality, which are exploited in all benchmarks. Finally, we evaluate the network on in-vivo measured catheter data.

## 4.1. Numerical experiments

In this section, we present the numerical results obtained with the proposed method. During the training process, we optimize both the network parameters $\theta$ and the latent code representations $\mathbf{z}_i$ of the training data. The network is fit to 4 surfaces, namely the epi- and endocardium of the right and left ventricle, and the associated hyperparameters $\sigma$ and $\alpha$ are optimized for the best performance of the reconstruction in the shape completion process. For the loss term, we thus obtained $1/\sigma^2 \approx 1.8 \times 10^{-7}$ and $\alpha = 1.9 \times 10^{-6}$. The network is trained with the Adam optimizer (Kingma and Ba, 2015) for 3000 epochs with a learning rate of 0.005 which is decreased twice with the factor of 0.2 after 2700 and 2900 epochs. For the mesh generation, we compute the signed distance function on a $128^3$ grid on the bounding box of all training points and reconstruct the zero-level set using the contour filter of PyVista (Sullivan and Kaszynski, 2019), based on marching cubes. The final reconstruction quality of the training models is depicted in Figure 1 (right).

We test the regularity of our network with respect to the latent space input with inter- and extrapolation between two latent codes of the training dataset. The shapes of the reconstructed hearts change uniformly between every inter- and extrapolation point and generate meaningful results (see appendix Figure 4). Points that are drawn from the prior distribution with covariance $\sigma^2 I$ produce meaningful heart geometries (see appendix Figure 5). With our model, we are able to learn multiple implicit surfaces at the same time and encode them in a joint latent space representation. Therefore, it is easily possible to calculate the latent code during inference time based on an arbitrary subset of surface points. We test this by sampling sparse point clouds on the endocardium of the left ventricles from the test set and optimizing the latent code according to (2), still providing us with a full 4-chamber biventricular shape. Not only is our method capable of a close reconstruction of the endocardium of the left ventricle, but it can also predict possible shapes of the other three surfaces. We provide visual results for a reconstruction from 50 lv endocardial points in Figure 2 and Figure 6. Note however that the accuracy of the unmeasured right ventricular shape significantly decreases. Numerical results of the reconstruction quality on the endocardium of the left ventricle can be found in Section 4.2.

## 4.2. Validation

Our method is compared with Points2Surf (Erler et al., 2020), Point2Mesh (Hanocka et al., 2020), and a variant of our network without Lipschitz regularization. In all experiments, only the endocardium of the left ventricle is considered. All methods are compared on point clouds with different cardinalities $n$, ranging from very sparse point clouds with $n = 50$ to relatively dense ones with $n = 2000$. Additionally, the coordinates of the points are perturbed with noise drawn from a Gaussian normal distribution with zero mean and fixed covariance $\xi^2 I$. For each number of points $n$ and level of noise $\xi$ we test the methods on the same 44-point clouds sampled from the meshes of the data set (for Points2Surf we apply a pre-trained model (Erler et al., 2020)). In the appendix we additionally provide a comparison to a SSM (see Appendix C). To test our method on all 44 meshes of the dataset we selected 11 disjoint test sets and train a different network on each remaining training sets. For both our unregularized ($\alpha = 0$) and regularized DeepSDF we take $\beta = \max(1, C_s \xi^2)$ and minimize the objective in (2) using the Adam optimizer (Kingma and Ba, 2015) with

| | $\xi$ | $n$ | Points2Surf | Point2Mesh | ours* | ours |
|---|---|---|---|---|---|---|
| | | 50 | - | 1.74±0.41 | *1.2*±0.91 | **0.75**±0.18 |
| | 0 | 200 | - | *0.68*±0.11 | 0.72±0.54 | **0.49**±0.09 |
| | | 500 | 1.66±0.32 | *0.57*±0.08 | 0.70±0.57 | **0.45**±0.09 |
| Chamfer Distance | | 50 | - | 2.33±0.54 | *1.65*±0.87 | **1.38**±0.28 |
| | 2 | 200 | - | *1.16*±0.13 | 1.16±0.57 | **0.96**±0.17 |
| | | 500 | 1.65±0.14 | *0.97*±0.10 | *0.97*±0.33 | **0.96**±0.11 |
| | | 50 | - | 3.50±0.49 | *2.6*±0.98 | **2.39**±0.48 |
| | 5 | 200 | - | 3.17±0.62 | *1.87*±0.39 | **1.84**±0.27 |
| | | 500 | 2.82±0.31 | 3.85±1.15 | *2.32*±1.00 | **1.99**±0.33 |

**Figure 2:** Left: mean and standard deviation of the Chamfer distance (CD) for different numbers of input points $n$ and different levels of noise $\xi$ (lowest value in **bold magenta**, second lowest value in *teal and italic*; short version of Table 1 in the appendix). The asterisk denotes the non Lipschitz regularized version of our network. Right: reconstruction of both endocardia from points on the LV endocardium (gt mesh in grey, reconstruction color-coded with implicit distance).

a learning rate of $10^{-2}$ for $50\,000$ epochs. In our unregularized network we used standard linear layers without any regularization and changed the activation functions to ReLu. For the unregularized layers this seems to improve the performance drastically, whereas for the regularized case we found that the choice of activation function did not influence the results too much. The results for $n = 50$ on four meshes computed with our method are depicted in Figure 7 in the appendix. For the comparison, we evaluated the performance in terms of *L2-Chamfer-distance* (CD), which can be seen in Figure 2. For two point clouds $\mathbf{X}$ and $\mathbf{Y}$ the CD is given as $d_{\mathrm{CD}}(\mathbf{X}, \mathbf{Y}) = \frac{1}{|\mathbf{X}|} \sum_{x \in \mathbf{X}} \min_{y \in \mathbf{Y}} \|x - y\|_2 + \frac{1}{|\mathbf{Y}|} \sum_{y \in \mathbf{Y}} \min_{x \in \mathbf{X}} \|x - y\|_2$. In appendix B and Table 1 further comparisons in terms of *Hausdorff-distance* (HD), and *Large deformation diffeomorphic metric mapping* (LDDMM) with their respective definitions are provided. For every test case, we state the mean and standard deviation across the four meshes of the test set. Note that Points2Surf did not converge for sparse point clouds ($n = 50$ and $n = 200$). We thus omitted the associated results in the table. Our method performs particularly well on sparse point clouds compared to the other methods. The obtained meshes are qualitatively similar to the ground truth meshes and geometric features like the curvature can be recovered properly (see appendix Figure 8).

## 4.3. Inference from electroanatomical mapping data

Electro-anatomical mapping (EAM) is a common routine for patients undergoing catheter ablation. As the catheter is inserted into the heart, it continuously builds a three-dimensional point cloud that is triangulated into a surface model. Our EAM data consists of a set of points located on the endocardial surface of the left ventricle. The point cloud is usually sparse and unevenly distributed, but commonly allows for a reasonable estimate of endocardial geometry. The exact data acquisition process is described in detail in appendix E. The

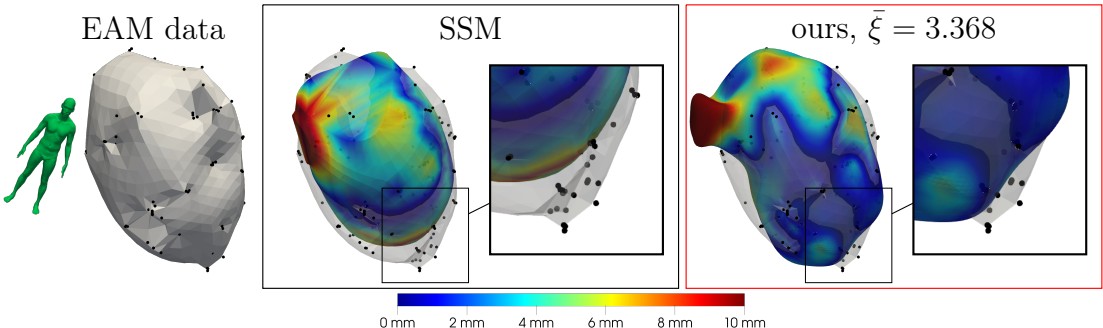

**Figure 3:** Reconstruction of the LV endocardium from the EAM data. We compare the SSM reconstruction (left panel) to our approach for an optimal noise level estimation of $\bar{\sigma} = 3.368$ mm (right panel). We also report the absolute distance from the geometry provided by the EAM system.

overall LV shape is well-approximated in most regions except for the apex, as can be seen in Figure 3 and Figure 9, and includes the outflow aortic tract due to the learned modality. Compared to the LV surface obtained from the EAM system, our shape is much smoother and is even defined in regions where data is missing. Since the underlying noise structure is unknown, we estimate the noise variance iteratively as described in (3). For the EAM data, we obtain an estimate of 3.4 mm, consistent with the size of the electrode's tip $(2 - 4$ mm). A table containing the iterates of the estimator can be found in Table 3 in the appendix.

## 5. Discussion and outlook

We have presented a novel method to represent cardiac anatomy based on signed-distance functions. The quantitative comparison with other methods shows that our approach can reconstruct the shape of hearts on a state-of-the-art level, especially for sparse and noisy data. In contrast to the methods we compared against, our approach does not require knowledge of the normals at each point of the point cloud. This knowledge however could be incorporated at the shape completion stage by fitting the normalized gradient of the SDF against the given normals.

The requirement of point clouds is a mixed blessing, as it requires the construction of a point cloud from images through a learned method, or by segmenting the image stack and using its surface. On the other hand, enables the algorithm to operate on different modalities. Additionally, the algorithms expects the point clouds in a canonical pose (CT-based), which however could be automated using rigid point registration methods Jian and Vemuri (2010). Learning multiple SDFs with one network allows the generation of shapes based on input data from different chambers of the heart or a combination of multiple surfaces.

While the present work only encodes bi-ventricular surfaces, the method itself can be extended to encode additional chambers and shapes, such as the atria or aorta. This method has important implications in cardiac modeling, digital twinning for precision medicine, and the creation of virtual cohort of patients. We plan to further extend it to time-dependent shape models and shape uncertainty quantification (Gander et al., 2021).

## 6. Acknowledgements

This work was supported by the Swiss National Science Fund [Cardiotwin, Weave/Lead Agency, Project number 214817], by the Deutsche Forschungsgemeinschaft (DFG, German Research Foundation) [EXC-2047/1-390685813] and [EXC2151-390873048], PRIN-PNRR [project no. P2022N5ZNP], and Swiss National Supercomputing Centre [production grant no. s1074].

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

## Appendix A. Dataset

Based on Rodero et al. (2021); Strocchi et al. (2020b), we built a public shape library of watertight 3D shapes for endo-/epicardium of the left/right ventricles (4 shapes in total). Note that the dataset is composed of 20 patients that are diagnosed as healthy (no cardiac conditions detected) Rodero et al. (2021) and 24 patients are diseased with various heart failures recruited for cardiac resynchronization therapy (CRT) upgrade Strocchi et al. (2020b,a). The generated shape data is available as a Zenodo record (Grandits et al., 2024).

### A.1. Surface mesh generation

As a basis for our meshes, we used the publicly available data described in Rodero et al. (2021); Strocchi et al. (2020b). Our dataset was produced by applying the following procedure to each of the meshes: First, after loading the data, we localize the left ventricular apex using the available universal ventricular coordinates (UVCs, (Bayer et al., 2018)) and use it as the Cartesian origin for each of the meshes. Next, we extract the left/right ventricular (LV/RV) endo-/epicardia by using the available surface tags. The provided meshes encode the walls and valves into separate tags, from which we extracted the surfaces. We then identify all points of the wall that touch the associated valve (e.g. LV wall with mitral and aortic valves). To identify all reachable points, we compute an eikonal solution using (Grandits, 2021) by placing an initial point on the inside of the wall (closest to the blood pool) with minimal velocity across points that touch any of the valves. The wall is then separated into epi- and endocardium by applying a thresholding filter on the solution. Finally, we remove non-manifold parts, recalculate the inside/outside orientation and close the surface to receive a watertight, proper manifold surface for the endo- and epicardia separately. Note that all steps exploit VTK (Schroeder et al., 2006) through PyVista (Sullivan and Kaszynski, 2019).

### A.2. Surface sampling

The training data was then generated by sampling $3,000$ surface points (surface samples) and additional $1,000$ points, displaced in normal direction randomly (uniform) by up to 30mm (band samples) per surface, i.e. $16,000$ points per patient. This point cloud was sampled with a curvature-based weighting to create more samples around features of interest (e.g. apex, valves). Specifically, we generated the dataset samples using face-weights $w$ following Gao et al. (2019, Eq. 2.6) that are defined as follows: consider a 2-dimensional compact surface $M$, isometrically embedded in $\mathbb{R}^3$, with its Gaussian curvature $\kappa$ and mean curvature $\eta$. The weighting function is then defined as

$$w_{\lambda,\rho}(\mathbf{x}) = \frac{\lambda|\kappa(\mathbf{x})|^\rho}{\int_M |\kappa(\xi)|^\rho \, \mathrm{dvol}_M(\xi)} + \frac{(1-\lambda)|\eta(\mathbf{x})|^\rho}{\int_M |\eta(\xi)|^\rho \, \mathrm{dvol}_M(\xi)}. \tag{4}$$

In our experiments, we used $w_{0.1,0.75}$ as the curvature weight, purely by visual inspection. We used the library trimesh (Dawson-Haggerty et al., 2009) for sampling and computing $\kappa$ and $\eta$.

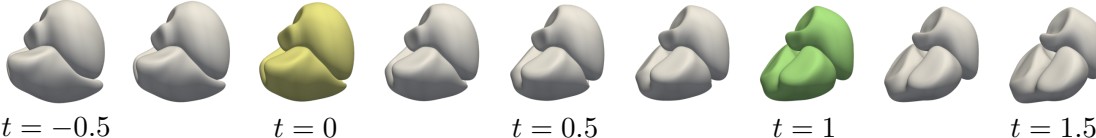

$t = -0.5$  $t = 0$  $t = 0.5$  $t = 1$  $t = 1.5$

**Figure 4:** The figure shows the generation of heart models via interpolation and extrapolation of latent code vectors. In detail, we take two latent code representations $\mathbf{z}_1, \mathbf{z}_2$ from the training dataset and decode the linear combinations $t\mathbf{z}_1 + (1-t)\mathbf{z}_2$. We present the reconstructed endocardia.

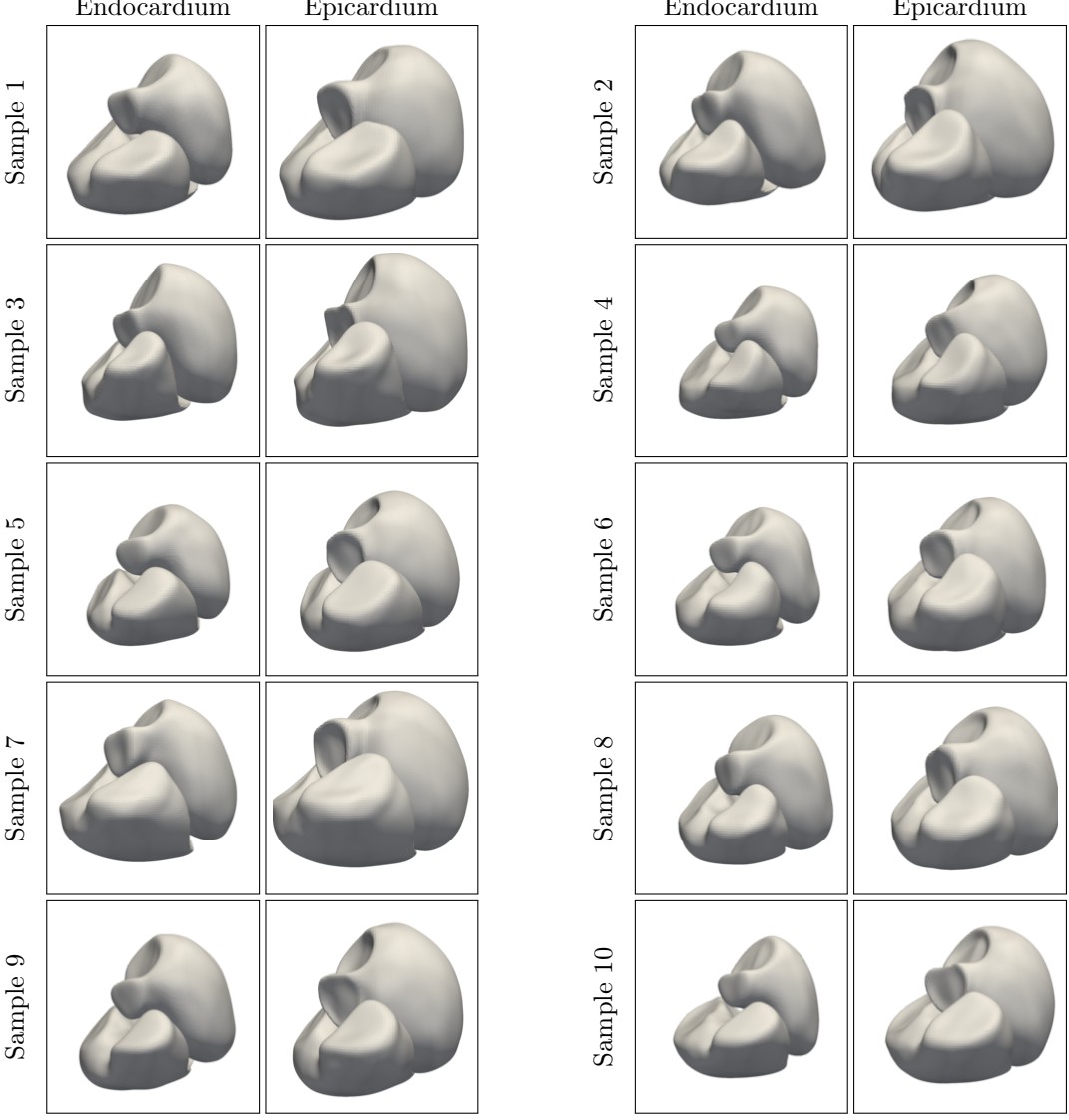

**Figure 5:** We reconstruct points in the latent space that are sampled from a zero-mean Gaussian distribution with covariance $\sigma^2 I$.

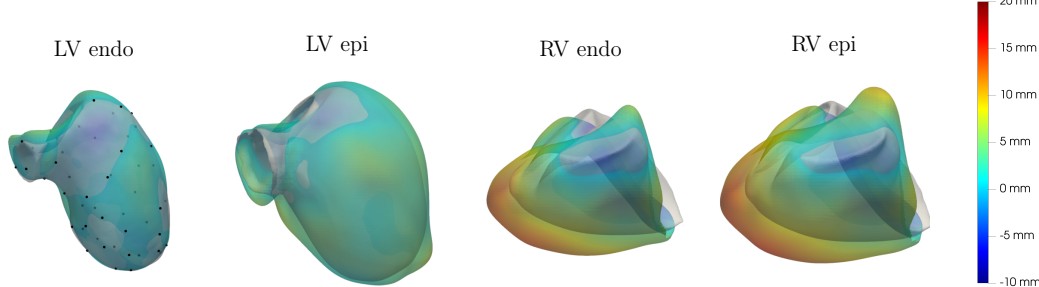

**Figure 6:** We reconstruct all four surfaces from 50 points on the endocardium of the left ventricle. The reconstructed meshes are color-coded with the mesh distance, i.e. the distance to the nearest point on the ground truth mesh in mm.

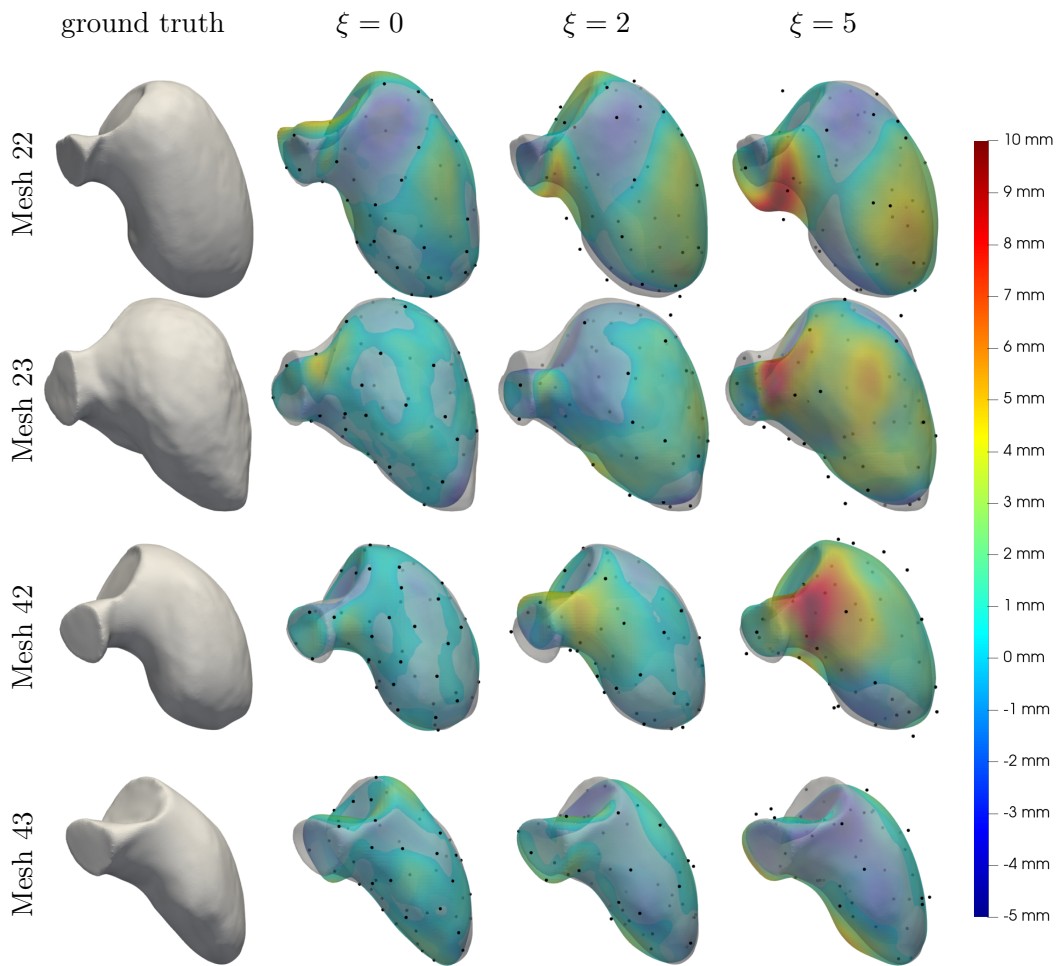

**Figure 7:** Reconstruction quality of the left ventricle from 50 points with different levels of noise. We color-coded the implicit distance to the ground truth mesh.

## Appendix B. Metrics

To quantify the quality of reconstructed meshes we use the *L2-Chamfer-distance* (CD), the *Hausdorff-distance* (HD), and a *Large deformation diffeomorphic metric mapping* (LDDMM) loss terms between the reconstructed mesh and the ground truth mesh. We calculate the L2-Chamfer distance and the Hausdorff distance by sampling $50\,000$ points on each mesh. For two point clouds $\mathbf{X}$ and $\mathbf{Y}$ the CD is given as

$$d_{\mathrm{CD}}(\mathbf{X}, \mathbf{Y}) = \frac{1}{|\mathbf{X}|} \sum_{x \in \mathbf{X}} \min_{y \in \mathbf{Y}} \|x - y\|_2 + \frac{1}{|\mathbf{Y}|} \sum_{y \in \mathbf{Y}} \min_{x \in \mathbf{X}} \|x - y\|_2.$$

The Hausdorff distance (HD) is defined as

$$d_{\mathrm{HD}}(\mathbf{X}, \mathbf{Y}) = \max\{\max_{x \in \mathbf{X}} \min_{y \in \mathbf{Y}} \|x - y\|_2, \max_{y \in \mathbf{Y}} \min_{x \in \mathbf{X}} \|x - y\|_2\}.$$

With the LDDMM loss, we measure how well the reconstructed mesh $\mathcal{M}_r$ can be registered to the original ground truth mesh $\mathcal{M}_{gt}$. To obtain numerically comparable results across the different methods we first remesh the results to obtain meshes with the same resolution (for this task, we use the Blender software package (Blender Online Community, 2018) with a voxel resolution of 0.9 mm). To obtain the LDDMM loss, we calculate the center points $c_F$, the normals $n_F$ and the area $A_F$ of every face $F$ from the set of faces $\mathcal{F}_r$ and $\mathcal{F}_{gt}$, respectively. For $\gamma = 1$ let

$$C(\mathcal{F}_1, \mathcal{F}_2) = \sum_{F_1 \in \mathcal{F}_1} \sum_{F_2 \in \mathcal{F}_2} e^{-\gamma \|c_{F_1} - c_{F_2}\|_2^2} \langle n_{F_1}, n_{F_2} \rangle A_{F_1} A_{F_2}.$$

Then, the LDDMM loss is defined as

$$d_L(\mathcal{M}_{gt}, \mathcal{M}_r) = C(\mathcal{F}_{gt}, \mathcal{F}_{gt}) + C(\mathcal{F}_r, \mathcal{F}_r) - 2C(\mathcal{F}_{gt}, \mathcal{F}_r).$$

## Appendix C. Fitting the Statistical Shape Model

The statistical shape model describes a heart shape as a variation of a mean shape $\mu$ in different directions (modes). We used the publicly available SSM from the cardiac atlas project that is based on 630 healthy Biobank reference patients, further described in Petersen et al. (2017). A mean point cloud $\mathbf{X} \in \mathbb{R}^{N \times 3}$ of $N$ points together with 200 eigenmodes $\mathbf{V}_i \in \mathbb{R}^{N \times 3}$ for $i = 1, \ldots, 200$ and corresponding eigenvalues $\lambda \in \mathbb{R}^{200}$ are provided, which is restricted to the subset of points corresponding to the endocardium of the left ventricle. For a given sample with weights $\alpha \in \mathbb{R}^{200}$ and a spatial offset $\mathbf{b} \in \mathbb{R}^3$, we obtain a representation of the resulting point cloud $C$ as $\mathbf{C} = \mathbf{X} + \mathbf{1}_N \mathbf{b}^\top + \sum_{i=1}^{200} \alpha_i \lambda_i \mathbf{V}_i$. For a given target point cloud $\mathbf{Y}$ we use this model to optimize the asymmetric Chamfer distance $d_{\mathrm{aCD}}(\mathbf{Y}, \mathbf{C}) = \frac{1}{|\mathbf{Y}|} \sum_{y \in \mathbf{Y}} \min_{x \in \mathbf{C}} \|x - y\|_2$ w.r.t. the weights $\alpha$ and the spatial offset $\mathbf{b}$. Additionally, we use a standard $\ell_2$ loss term to obtain the final objective function

$$\mathcal{J}_{\mathbf{Y}}(\alpha, \mathbf{b}) = d_{\mathrm{aCD}}\left(\mathbf{Y}, \mathbf{X} + \mathbf{1}_N \mathbf{b}^\top + \sum_{i=1}^{200} \alpha_i \lambda_i \mathbf{V}_i\right) + \beta \|\alpha\|_2.$$

This loss function is optimized for $5\,000$ epochs with the Adam optimizer (Kingma and Ba, 2015), with a learning rate of 0.005.

| | ξ | n | Points2Surf | Point2Mesh | SSM | ours* | ours |
|---|---|---|---|---|---|---|---|
| **Chamfer Distance** | 0 | 50 | - | 1.74±0.41 | 3.58±1.04 | *1.20*±0.91 | **0.75**±0.18 |
| | | 200 | - | *0.68*±0.11 | 2.73±0.77 | 0.72±0.54 | **0.49**±0.09 |
| | | 500 | 1.66±0.32 | *0.57*±0.08 | 2.34±0.76 | 0.70±0.57 | **0.45**±0.09 |
| | | 2000 | 0.79±0.17 | 0.57±0.11 | 1.87±0.83 | *0.54*±0.32 | **0.43**±0.07 |
| | 2 | 50 | - | 2.33±0.54 | 3.60±1.38 | *1.65*±0.87 | **1.38**±0.28 |
| | | 200 | - | *1.16*±0.13 | 2.79±0.82 | *1.16*±0.57 | **0.96**±0.17 |
| | | 500 | 1.65±0.14 | *0.97*±0.10 | 2.33±0.68 | *0.97*±0.33 | **0.96**±0.11 |
| | | 2000 | 1.85±0.42 | **0.80**±0.13 | 1.88±0.86 | *1.11*±0.78 | 1.48±0.52 |
| | 5 | 50 | - | 3.50±0.49 | 3.79±1.34 | *2.60*±0.98 | **2.39**±0.48 |
| | | 200 | - | 3.17±0.62 | 3.00±0.88 | *1.87*±0.39 | **1.84**±0.27 |
| | | 500 | 2.82±0.31 | 3.85±1.15 | 2.49±0.70 | *2.32*±1.00 | **1.99**±0.33 |
| | | 2000 | 3.22±0.32 | 5.07±0.94 | **2.08**±0.92 | 4.24±1.50 | *3.10*±0.79 |
| **Hausdorff Distance** | 0 | 50 | - | 10.86±3.45 | 11.96±3.69 | *7.54*±7.25 | **4.40**±1.82 |
| | | 200 | - | *4.23*±1.18 | 9.93±3.88 | 4.52±5.79 | **2.57**±0.92 |
| | | 500 | 9.88±3.84 | **2.45**±0.55 | 9.81±5.45 | *4.14*±5.01 | 4.43±15.03 |
| | | 2000 | 6.30±0.73 | **1.66**±1.11 | 11.29±9.16 | 2.99±2.91 | *2.00*±0.79 |
| | 2 | 50 | - | 12.86±3.99 | 12.27±4.58 | *7.75*±4.40 | **6.01**±2.54 |
| | | 200 | - | *5.82*±1.17 | 10.27±4.04 | 6.07±4.20 | **4.36**±1.32 |
| | | 500 | 10.29±1.02 | *4.53*±1.12 | 9.69±5.07 | 5.31±5.27 | **4.41**±1.14 |
| | | 2000 | 10.04±0.73 | **5.15**±2.06 | 11.52±9.21 | *7.03*±5.38 | 10.82±5.50 |
| | 5 | 50 | - | 16.13±3.09 | 12.61±4.10 | *10.58*±5.41 | **8.72**±2.22 |
| | | 200 | - | 14.94±2.85 | 11.25±4.35 | *8.34*±2.56 | **7.78**±1.91 |
| | | 500 | 18.68±2.10 | 15.83±2.56 | 10.78±5.02 | *10.43*±4.96 | **8.66**±2.12 |
| | | 2000 | 19.77±1.46 | 17.95±3.34 | **13.20**±9.14 | 19.64±6.48 | *15.29*±4.12 |
| **LDDMM Loss (×10³)** | 0 | 50 | - | 73.5±24.97 | 99.21±31.57 | *53.39*±25.22 | **35.01**±14.96 |
| | | 200 | - | 34.82±14.33 | 93.53±30.00 | *29.81*±20.54 | **17.20**±8.81 |
| | | 500 | 75.51±25.72 | *26.01*±11.80 | 89.67±29.86 | 27.16±25.30 | **12.42**±6.49 |
| | | 2000 | *14.07*±7.05 | 25.16±12.85 | 80.41±26.87 | 18.93±15.98 | **11.84**±7.21 |
| | 2 | 50 | - | 88.64±26.12 | 99.05±31.77 | *72.63*±24.63 | **68.98**±22.57 |
| | | 200 | - | 65.05±19.99 | 94.76±30.62 | *57.96*±21.68 | **51.56**±17.79 |
| | | 500 | 78.18±23.83 | 55.72±18.50 | 90.19±30.38 | **50.98**±16.10 | *52.68*±14.37 |
| | | 2000 | **38.39**±12.96 | *43.59*±15.17 | 80.19±26.93 | 51.79±19.65 | 63.63±13.81 |
| | 5 | 50 | - | 109.14±27.98 | 101.57±37.27 | *91.93*±26.39 | **90.50**±24.21 |
| | | 200 | - | 108.31±26.01 | 97.98±34.19 | *82.23*±21.85 | **82.23**±21.26 |
| | | 500 | 89.19±25.25 | 114.69±18.64 | 92.37±32.41 | *88.77*±19.81 | **84.5**±21.61 |
| | | 2000 | **76.54**±18.93 | 131.00±27.89 | *83.01*±26.17 | 122.25±37.38 | 98.92±20.20 |

**Table 1:** Mean and standard deviation of the Chamfer distance (CD), the Hausdorff distance, and the LDDMM-loss for different numbers of input points $n$ and different levels of noise $\xi$. For every test case, the lowest value is printed in bold and the second lowest value is printed in italic. The asterisk denotes the non Lipschitz regularized version of our network.

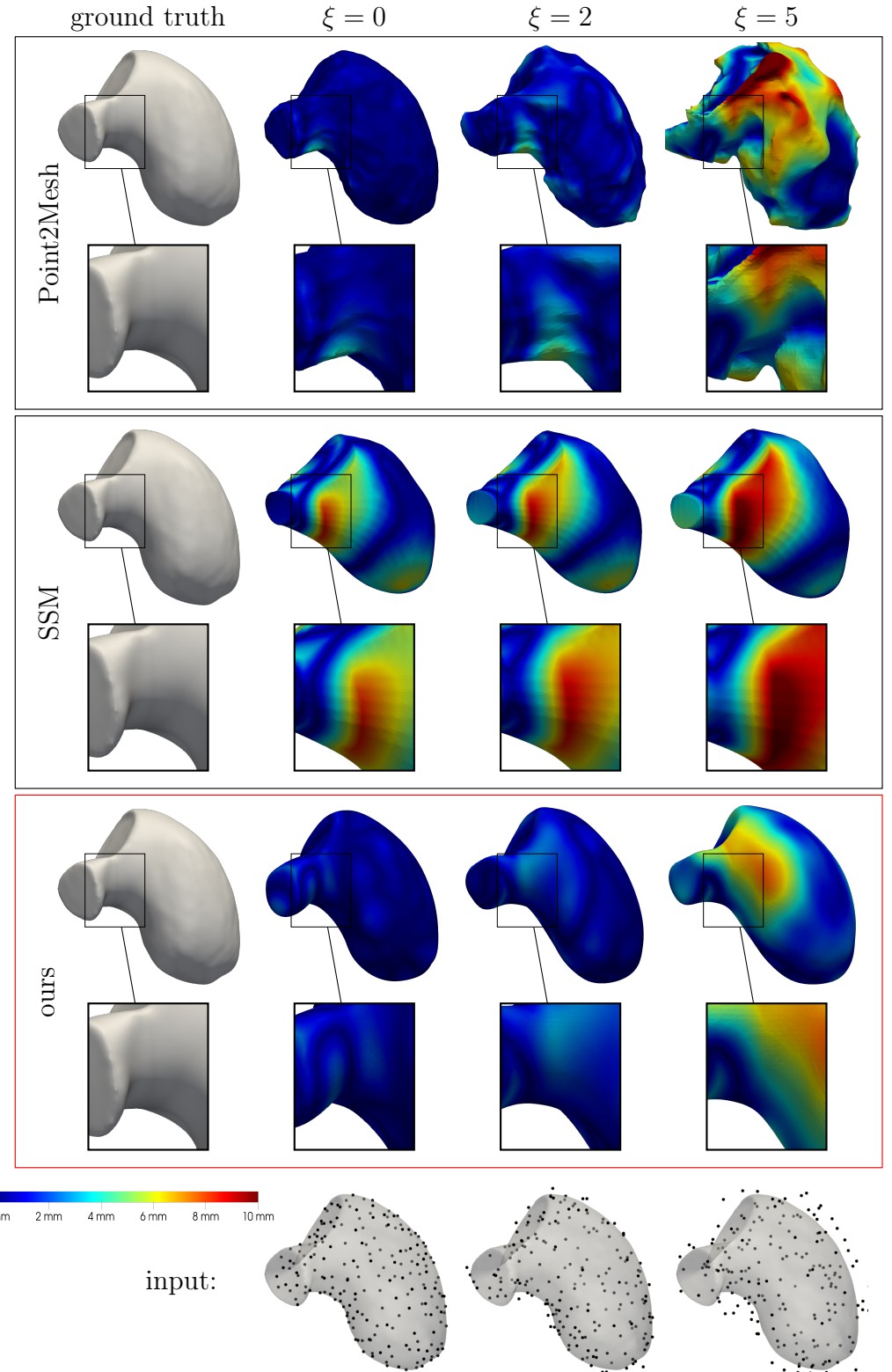

**Figure 8:** Mesh quality for different methods on mesh 43. We depict the result for 200 points. We color-coded the absolute distance to the ground truth mesh on the remeshed results.

## Appendix D. Multichamber Experiments

In this section, we investigate the advantages of jointly encoding of left and right endocardial shapes in comparison to independently reconstructing the two shapes. We then compare the average chamfer distances across both endocardia for the case of a separate reconstruction and a joint reconstruction for four test meshes. Additionally, we perform the same experiment on partially observable regions, where the points on the left ventricle are only sampled from its top half (ventricular base) and the points of the right ventricle are sampled from its lower half (ventricular apex). The results of both experiments can be found in Table 2. For the case of full data availability, the joint reconstruction seems to be advantageous only for the case of very sparse data ($n = 10$), but for the cases where only partial data is available, the joint reconstruction improves the resulting reconstruction in all but two cases.

| | | full data | | partial data | |
|---|---|---|---|---|---|
| $\sigma$ | $n$ | $CD_{\text{separate}}$ | $CD_{\text{joint}}$ | $CD_{\text{separate}}$ | $CD_{\text{joint}}$ |
| 0 | 10 | 2.71 | **2.47** | 3.31 | **2.50** |
| | 20 | **1.57** | 1.86 | 2.76 | **2.38** |
| | 30 | **1.43** | 1.42 | **2.34** | 2.77 |
| | 40 | **1.17** | 1.31 | **2.41** | 2.46 |
| 2 | 10 | 3.13 | **3.06** | 4.16 | **2.81** |
| | 20 | 2.43 | **2.39** | 3.24 | **2.72** |
| | 30 | 2.01 | 2.01 | 3.35 | **2.50** |
| | 40 | **1.83** | 2.12 | 3.24 | **2.82** |
| 5 | 10 | 3.63 | **3.49** | 3.95 | **3.92** |
| | 20 | **3.42** | 3.47 | 4.51 | **4.12** |
| | 30 | **3.04** | 3.25 | 3.76 | **3.70** |
| | 40 | **2.86** | 2.87 | 3.89 | **3.71** |

**Table 2:** Average reconstruction quality per surface for inference based on $n$ points on the LV Endocardium only.

## Appendix E. Obtaining the EAM data

The EAM data has been acquired with the NOGA-XP system (Biologic Delivery Systems, Biosense Webster) equipped with a conventional 7-Fr deflectable-tip mapping catheter (NAVI-STAR, Biosense Webster). Spatial positions of the tip of the catheter were acquired at 100 Hz, and aligned in time using the automatically detected R-peak of the 12-lead ECG. Points were accepted by the system according to a set of criteria for catheter stability and signal quality. The institutional review board approved the study protocol, and all patients gave written and oral informed consent for the investigation (the study is compliant with the Declaration of Helsinki). Further information on the study has been previously reported Maffessanti et al. (2020); Pezzuto et al. (2021). We pre-processed the data by applying a $-90°$ rotation about the $X$-axis (NOGA to LPS-MRI coordinate system), fol-

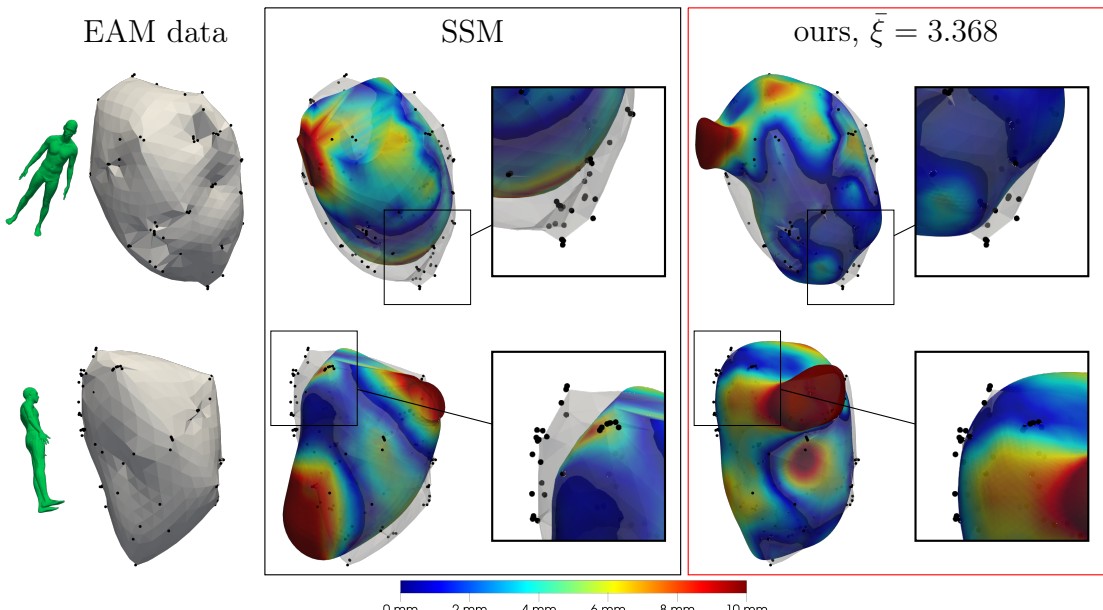

**Figure 9:** Reconstruction of the LV endocardium from the EAM data. We compare the SSM reconstruction (left panel) to our approach for an optimal noise level estimation of $\bar{\sigma} = 3.368$ mm (right panel). We also report the absolute distance from the geometry provided by the EAM system.

lowed by a translation to align the LV epicardial apex with the origin. Note that only an approximate alignment is possible since the epicardium is not present in the EAM data.

## Appendix F. Estimation of noise level

In Table Table 3 we present the numerical convergence of the noise estimator $\bar{\xi}_i$ for the first six steps on the EAM data (c.f. Section 4.3). Additionally, the performance of this method is tested for a synthetic point cloud (Heart 43, $n = 500$, $\xi = 5$), where the correct noise level is obtained.

| step | synthetic | | EAM data | |
|---|---|---|---|---|
| $\bar{\xi}_0$ | 0 | 15 | 0 | 15 |
| $\bar{\xi}_1$ | 3.159 | 4.882 | 1.499 | 4.672 |
| $\bar{\xi}_2$ | 4.878 | 5.063 | 3.184 | 3.686 |
| $\bar{\xi}_3$ | 5.063 | 5.060 | 3.344 | 3.398 |
| $\bar{\xi}_4$ | 5.060 | 5.060 | 3.797 | 3.367 |
| $\bar{\xi}_5$ | 5.060 | 5.060 | 3.410 | 3.364 |
| $\bar{\xi}_6$ | 5.060 | 5.060 | 3.368 | 3.364 |

**Table 3:** Iteration of the variance estimation for the synthetic case (Heart 43, $n = 500$, $\xi = 5$) and the EAM data.

