# OpenReview forum: "Shape of my heart: Cardiac models through learned signed distance functions"
_MIDL.io/2024/Conference — MIDL 2024 Poster_

### Official Review · Reviewer_7Bgm · 2024-02-22

**Confidence:** 5
**Preliminary Rating:** 1
**Final Rating:** 1

**Summary:**

The authors present a method for cardiac shape generation using implicit neural representations. The method is very similar to the one presented in Sander et al. 2023, but rather than completing shapes in sparsely sampled CMRI, the authors propose to complete shapes in sparsely sampled electro-anatomical maps. They extend the method by utilizing multiple output heads for four classes (i.e. epi- and endocardium of both ventricles), rather than a single ventricle per MLP. Additionally, they add a Lipschitz regularization term to improve the plausibility of reconstructions from randomly sampled latent codes, as proposed in Liu et al. 2022.

**Strengths:**

The descriptions is the paper are generally thorough and the proposed use of the Lipschitz normalization term seems sensible for this application. Additionally, the authors mostly used public data to evaluate the method.

**Weaknesses:**

The manuscript overstates its novelty. The authors compare their work with outdated methods, which misrepresents the benefit of the presented method. Additionally, while Sander et al. 2023 is cited in the related work section, the rest of the manuscript is written as if this (very similar) work does not exist.

The additions compared to Sander et al. 2023 are minor, and these additions are not properly compared with the baseline. The primary innovation over the Sander et al. is the Lipschitz regularization term, but the authors provide no comparison or evaluation regarding the effects and benefits of this regularization term.

While the authors show that the method can generate plausible inter- and extrapolations from the latent vectors of two shapes, the motivation for why this is relevant is not clear from the manuscript.

The method is only evaluated on four patients.

Not all claims are substantiated with experimental results. While the authors state "[our method] can also predict the shape of the other three surfaces", they do not provide numerical results for the errors in any of these reconstructions.

The discussion is not very well written. It does not discuss the implications or effects of the proposed additions compared to previously published methods.

**Detailed Comments:**

See: Questions To Address In The Rebuttal

**Justification Of Final Rating:**

The new multi-chamber comparison unfortunately has the appearance of cherry-picked and misrepresented results. In all of the original experiments, the authors show the results for 50, 200, 500 and 2000 samples. In appendix D, the table conspicuously stops at 40. The authors admit that the joint reconstruction seems to be advantageous only for the case of very sparse data (n = 10), which does not seem to be a relevant use-case (as the lower bound in the rest of the manuscript is 50). These results show that for the use cases considered in the rest of the manuscript, the multi-class approach is actually worse than the single-class version proposed in Sander et al. 2023.

The updated manuscript still includes the sentence "We demonstrate that this approach is also capable of reconstructing anatomical models from partial data, such as point clouds from a single ventricle" in the abstract, which is misleading. The authors have not added the numerical results for the other classes in this experimental setting, and the result they show in Figure 6 implies this is _not_ possible with any reasonable expectation of accuracy. Additionally, the manuscript still includes the same second and third paragraph of the introduction, which imply that the current work is the first to propose cardiac shape generation with implicit SDFs. This is only later toned down by the sentence "The proposed method shares some similarities with the very recent work by Sander et al. (2023)", which is not an adequate characterization of the similarity of the methods. Especially as the results still label the version without Lipschitz regularization as __Ours (L)__, even though the authors claim this is their best effort at a reimplementation of Sander et al. in the OpenReview comments (_"In the view of the authors, the non-Lipschitz regularized DeepSDF is the closest achievable approximation to the paper by Sanders et al., ..."_). Furthermore, it seems like the authors still use the multi-class setting for this baseline, which from the results shown in Appendix D would perform worse than the "true" baseline as presented in the original work.

**Justification Of The Preliminary Rating:**

While presented as a novel method, this method is very similar to previously published work (i.e. J Sander et al. 2023). The presented method included two methodological differences, but the manuscript contains no comparison or ablation experiments to evaluate the benefits of these changes.

**Questions To Address In The Rebuttal:**

The authors should compare their work with modern baselines for this task, such as J Wang et al. 2021 (MICCAI), M Beetz et al. 2022 (PMLR) and certainly with J Sander et al. 2023 (CBM). Does the proposed method outperform these methods? If not, what is the benefit of the proposed method compared to these methods?

The paper should be rewritten such that it no longer overstates its own novelty compared to related prior work.

What is the added benefit of the proposed Lipschitz scaling regularization term? Does it actually help? This should be investigated.

The authors should evaluate on more patients using cross-validation, rather than only reporting metrics for four patients.

The authors claim the method can predict the shape of the other three surfaces if only one of the surfaces is available, but it seems like such a system could only reconstruct a hallucination that does not necessarily resemble the true structures. What are the errors for these reconstructions, numerically?

What are the implications of the proposed additions compared to previously published methods?

**Special Issue:**

No

---

> ### Author Response · Authors · 2024-03-16
> **Answer to the review**
>
> Thank you for carefully reading our manuscript and providing valuable feedback. We are confident to address your critique convincingly and highlighted our novelty and differences to the related methods.
>
> Point-by-point response to the questions to address in the rebuttal:
> * The authors should compare their work with modern baselines for this task, such as J Wang et al. 2021 (MICCAI), M Beetz et al. 2022 (PMLR) and certainly with J Sander et al. 2023 (CBM). Does the proposed method outperform these methods? If not, what is the benefit of the proposed method compared to these methods?
>   + Thank you for your comment. Indeed, we were aware of the highly interesting paper by J Sander et al. 2023 (CBM), which we carefully read. However, originally we decided to compare our method with the established and widespread methods for the task of point cloud reconstruction, namely Points2Surf and Point2Mesh.
>
>     In contrast to the paper by Sanders et al., we further extend existing DeepSDF methodologies by incorporating Lipschitz regularization as proposed by Liu et al., together with a multi-SDF estimation approach. In our work, we particularly focus on the case of low density and high noise point cloud recordings, as often found in electroanatomical mapping, in which these architectural changes show major improvements. In the revised version of the manuscript, we have now added several new numerical experiments to exactly examine the benefit of the Lipschitz constraint, which outperforms non-Lipschitz regularized DeepSDFs in most cases considered. In the view of the authors, the non-Lipschitz regularized DeepSDF is the closest achievable approximation to the paper by Sanders et al., given that the training data they used is not publicly available, thus showing the superiority of our approach against the method by Sanders in the considered scenarios.
>
>     Second, we advocate a multi-region approach, in which simultaneously four signed distance functions are estimated. For instance, the simultaneous estimation of the endocardia of the left and right ventricle can improve the overall reconstruction quality for sparse and/or partial data, as we now show in Appendix D. We have systematically analyzed this in a novel numerical experiment. All of the aforementioned differences and benefits of our method are now further outlined in the introduction.
>
>     We also wanted to thank you for bringing the works of Beetz et al. and Wang et al. to our attention. Due to time and space restrictions of this work, we decided to compare our method in the revised version only against an approximate version of Sanders et al.. Note that in this very same work, they already showed how their method is superior to the works by Beetz et al. and Wang et al., which by extension also applies to our method since we outperform the work of Sanders et al. in most cases.
>
> * The paper should be rewritten such that it no longer overstates its own novelty compared to related prior work.
>   + Thank you for this comment. As stated in the previous reply, there exists a clear novelty to current methods. We however completely agree that we did not accurately describe these novel points, which resulted in the impression that we overstated our novelty (this was by no means our intention). In the revised version, we focused on a more careful description of our real novelty in the introduction and related work section.
>
> * What is the added benefit of the proposed Lipschitz scaling regularization term? Does it actually help? This should be investigated.
>   + Please see our response to the first item.
>
> * The authors should evaluate on more patients using cross-validation, rather than only reporting metrics for four patients.
>   + You are completely right. In the revised version, we have now included the cross-validation on all 44 available patient data as requested.
>
> * The authors claim the method can predict the shape of the other three surfaces if only one of the surfaces is available, but it seems like such a system could only reconstruct a hallucination that does not necessarily resemble the true structures. What are the errors for these reconstructions, numerically?
>   + Thank you for bringing this to our attention. We see potential in the possibility to reconstruct (unobserved) RV shapes from purely LV measurements. Since it is likely that the RV shape is not uniquely defined by the LV, the observed errors are comparably high, but significantly decrease once measurement points at the RV are added (see Appendix D in the revised version). Nevertheless, we can reconstruct a possible/likely candidate for the other structures based on the joint prior distribution. We adapted our statement in the paper accordingly.
>
> * What are the implications of the proposed additions compared to previously published methods?
>   + We clarified the implication that our method has compared to previously published methods, especially Sanders et al..

---

### Official Review · Reviewer_BWNX · 2024-02-27

**Confidence:** 5
**Preliminary Rating:** 2
**Recommendation:** Poster
**Final Rating:** 5

**Summary:**

The authors present a novel alternative to traditional linear statistical shape models of the biventricular anatomy. In essence, the proposed method comprises an auto-decoder architecture that maps spatial coordinates and a shape encoding to a four-dimensional vector, which represents the signed distances to the left and right endocardium and epicardium, respectively. Despite training a shallow Lipschitz-regularized neural network for the auto-decoder architecture on only 36 patient geometries, the method outperforms the comparators (Points2Surf, Points2Mesh, and a linear SSM) in reconstructing the left endocardium from sparse and noisy point clouds as well as when reconstructing the surface of sparse electroanatomic maps.

**Strengths:**

- The main manuscript is very well written and it is accompanied by an extensive appendix, which complements the main manuscript well
- The quantitative evaluation is always supplemented with graphical illustrations of the error distribution
- The method is novel, interesting, and seems very valuable to the community
- The results are very compelling despite the limited amount of training shapes
- The authors intend to share the generated data upon completion

**Weaknesses:**

While I strongly believe that this manuscript is of high quality and that the proposed method has significant merits over traditional SSMs, I have several questions and concerns that I hope the authors could answer:

- From my point of view, one key limitation of the method was not properly addressed in the manuscript: the requirement that the biventricular shapes have to be in a canonical pose for training and inference. Could the authors please comment on this limitation and how such a system could be applied, e.g., when the target is in a rotated position?

- It is questionable to me whether the traditional SSM is a valid comparator. First of all, the SSM was trained on a different patient population, which is only comprising healthy patients. The test data, however, was extracted from a database of healthy and diseased patients. Could the authors please provide information on whether the data split held any diseased patients and please comment on how this could impact the results? In addition, to fit the SSM, only the shape coefficients and a translational offset are optimized. Could there be a rotational bias due to the construction of the SSM on a different dataset that is offsetting the errors?

- Moreover, it seems to me like the proposed method is either compared against a too simple approach (linear SSM) or against too sophisticated deep-learning-based approaches (Points2Surf and Point2Mesh). Similarly, no ablation study, e.g., on the impact of the Lipschitz-regularization, was run, which makes it difficult to assess whether the improved results are a result of the non-linear shape representation, the SDF representation, or something else. One possible additional comparator may be a non-linear SSM constructed from the training meshes via an autoencoder. In theory, this may also allow to compare the inter- and extrapolation quality of the learned shape encoding.

- One particular advantage of the proposed method is the adjustable sampling resolution. To reconstruct a mesh, a grid sampling pattern of 128^3 was used. It is not clear how finer or coarser sampling would impact the reconstructed mesh and the errors. In addition, it is not clear why a point-based approach would be preferred over, e.g., projecting only the (learned) shape encodings to a fixed four-dimensional grid of size 128^3 holding the signed distance maps.

- Even though the paper was overall easy to read and provided in most cases complete descriptions, I often found myself constantly jumping between the main manuscript and the appendix and I do not believe that the manuscript would be self-contained without the appendix. In addition, the discussion of the results is in comparison to the extent of this paper too short and shallow.

**Detailed Comments:**

- Figure 1: The authors mention in the text that the output of the 2nd hidden layer is concatenated with the spatial coordinate and the shape encoding. Could the authors please adapt the network illustration to reflect this?
- Figure 2: Even though one can clearly see that the shapes are well extra- and intrapolated, it may be even easier to comprehend in a small animation. Would it be possible for the authors to also publish a video of the deformation?
- Figure 3, table: It may be easier to read if not font types but colors are used to highlight the lowest and second lowest values. In addition, how do the errors behave in relation to the heart size?
- Figure 3, graphic: How many samples on the LV endocardium were used for the reconstruction?
- Methods, Lipschitz-regularized network explanation: Could the authors please clarify how the additional weight c_i is incorporated into the network forward computation (if at all)?
- What algorithm is used to reconstruct the surfaces from the SDFs?
- Inference on EAM data: What is the error when reconstructing the data with the SSM?
- Discussion: The authors state "The requirement of point clouds is a mixed blessing, as it requires an additional step to work on image input", but the inference in the first experiment was run over positions sampled on a grid, which, to my understanding, would allow direct application to image data. Could the authors please clarify what they mean?
- Could the authors please clarify whether the spatial coordinates were normalized?
- Idea for future research: It would be very interesting to see whether, when trained on large amount of healthy and diseased shapes, the learned shape encodings are also discriminative of the underlying diseases. In addition, it would be very valuable to not only study the error under sparse data, but also partial shape data, e.g., providing only data on the septum or free wall

**Justification Of Final Rating:**

I truly appreciate that the authors have responded in such detail to all of my questions and concerns (thank you!!).

From my point of view, the new experiments as well as the changes made to the manuscript are adequate and strengthen the paper. While there are obviously many different ways to further test the proposed method, I do understand that it is impossible to do everything in the short time frame of the rebuttal phase as well as to fit everything within the page limit. In my opinion, the proposed method has significant merits and the manuscript, in its revised form, would be of great interest to the MIDL community. I am happy to change my recommendation to acceptance.

**Justification Of The Preliminary Rating:**

Even though the manuscript has several strengths and it is noticeable that the authors put a lot of effort into it, there are, however, at the moment too many questions and concerns, which I hope to get answers on during the rebuttal.

**Questions To Address In The Rebuttal:**

Please see the comments under the weaknesses section and in the detailed comments. In addition, could the authors please comment on the following questions:

- Could the authors please explain whether and how this method could be extended to represent non-closed surfaces?

- In traditional SSMs one can retrieve the mean shape, do the authors see any way to retrieve something similar with the proposed approach?

- Another benefit of SSMs are point correspondences, which can be easily used to transfer data between geometries. How could the proposed method be used for such tasks?

- A shape encoding of 64 dimensions was chosen. Could the authors please comment on the impact of the latent dimension size onto the results?

- Appendix, A2: Could the authors please clarify why two point clouds were generated? In addition, could the authors please comment on the impact of only sampling within a 30mm distance to the shape? What happens if one would sample further away? Could this be an issue for very large or very small geometries?

---

> ### Author Response · Authors · 2024-03-16
> **Answer to the review 1/3**
>
> Thank you for your valuable feedback on our paper. We used the allocated rebuttal time to extend our manuscript both by experiments and further elaboration of our method. We hope that this revision will be able to address your concerns both in the revised paper and in the following answers.
>
> * One key limitation of the method was not properly addressed in the manuscript: the requirement that the biventricular shapes have to be in a canonical pose for training and inference. Could the authors please comment on this limitation and how such a system could be applied [...]?
>   + Thank you for this question. This is indeed a limitation of our method, as we require the left ventricular apex to be aligned and the heart to have the same pose as in a CT scan. Note however that rigid alignment methods offer reasonable, well-established methods to solve this problem. This is now shortly discussed in the Discussion.
>
> * It is questionable to me whether the traditional SSM is a valid comparator [...]. Could the authors please provide information on the data split and the impact on the results? [...] Could there be a rotational bias due to the construction of the SSM on a different dataset that is offsetting the errors?
>   + We decided to use SSM as a competitor because they are still popular in the cardiac community, see:
>      - https://doi.org/10.1109/EMBC48229.2022.9871327
>      - https://doi.org/10.1117/12.2550650
>      - https://doi.org/10.1371/journal.pcbi.1008851
>
>     The applied SSM should have the same spatial orientation as the meshes from the test set, therefore optimizing the shape coefficients and the translational offset should be sufficient.
>
>     Regarding the difference between healthy and diseased patients, we further investigated how well our approach performs when only training healthy patients (Rodero et al., 2021) while later inferring the shapes of diseased patients (Strocchi et al., 2020) and compared the results to a model that was trained on a subset of the diseased hearts. Both models performed worse than our original model but we found no significant performance difference to one another. The deterioration in terms of performance of the methods when only using healthy or diseased data to infer diseased hearts could be a consequence of the reduced number of training meshes.
>     Since, the training data of the SSM purely comes from cardiovascular magnetic resonance (CMR) studies, the reference orientation is the same as for the training and test data used in this study, making a rotational bias highly unlikely.
>
> * Moreover, it seems to me that the proposed method is either compared against a too simple approach (linear SSM) or against too sophisticated deep-learning-based approaches [...]. Similarly, no ablation study, e.g., on the impact of the Lipschitz-regularization, was run, which makes it difficult to assess whether the improved results are a result of the non-linear shape representation, the SDF representation, or something else. One possible additional comparator may be a non-linear SSM constructed from the training meshes via an autoencoder.[...]
>   + Thank you for your comment. We have selected Points2Surf and Point2Mesh due to their widespread use in recent baseline comparisons for shape reconstructions with point cloud data as single input (note that both methods were published in 2020). Moreover, we have now conducted the requested ablation study to validate the benefit of the Lipschitz constraint. A comparison against a non-linear SSM as you suggested would indeed be interesting but is unfortunately not realizable within this rebuttal period.
>
> * One particular advantage of the proposed method is the adjustable sampling resolution. [...] It is not clear how finer or coarser sampling would impact the reconstructed mesh and the errors. In addition, it is not clear why a point-based approach would be preferred [...].
>   + Thank you for this question. In several numerical experiments, we concluded that the 128^3 grid is sufficiently fine to fully represent the encoded shape. Experimentally, no benefit of finer meshes could be discovered, which, however, results in additional computation time. Note however that further usage of the reconstructed mesh could require a higher resolution model (e.g electro-mechanical, or bidomain simulations), which can be easily realized using our methodology.
> * Even though the paper was overall easy to read and provided in most cases complete descriptions, [...] I do not believe that the manuscript would be self-contained without the appendix. In addition, the discussion of the results [...] of this paper too short and shallow.
>   + Thank you for your comment. Indeed, we target a self-contained manuscript, which is challenging due to the page limit. We placed our primary emphasis on authoring a self-contained method section while striving to achieve maximum self-containedness in all subsequent sections. In the revised version, we slightly revised the text.

---

> ### Author Response · Authors · 2024-03-16
> **Answer to the review 2/3**
>
> A point-by-point response to the detailed comments:
>
> * Figure 1: The authors mention in the text that the output of the 2nd hidden layer is concatenated with the spatial coordinate and the shape encoding. Could the authors please adapt the network illustration to reflect this?
>   + Fixed
>
> * Figure 2: Even though one can clearly see that the shapes are well extra- and intrapolated, it may be even easier to comprehend in a small animation. Would it be possible for the authors to also publish a video of the deformation?
>   + Thank you for this idea. Unfortunately, we were not able to produce such a video until the first rebuttal deadline, but we are happy to provide it during the next week in the discussion period.
>
> * Figure 3, table: It may be easier to read if not font types but colors are used to highlight the lowest and second lowest values. In addition, how do the errors behave in relation to the heart size?
>   + Following your advice, we now included a color scheme in the table. We tested the correlation of heart sizes (LV volume) to reconstruction error (CD) for our methods in all experiments, but found no consistent correlation.
>
> * Figure 3, graphic: How many samples on the LV endocardium were used for the reconstruction?
>   + We used 50 points for the reconstruction, the information can now be found in the paper.
>
> * Methods, Lipschitz-regularized network explanation: Could the authors please clarify how the additional weight c_i is incorporated into the network forward computation (if at all)?
>   + Thank you for your question. We exploited an absolute row sum as the upper boundary for the Lipschitz constant. In every forward application of the network, we multiply every row of the weight matrix by a factor to ensure that its absolute sum is below c_i. For further details, we refer to (Liu et al., 2022).
>
> * What algorithm is used to reconstruct the surfaces from the SDFs?
>   + The meshes are generated using the contour filter of pyvista, which itself is based on marching cubes. This remark was now added to Sec. 4.1.
>
> * Inference on EAM data: What is the error when reconstructing the data with the SSM?
>   + We highlight that there is no ground truth available for the heart shapes of the EAM data. Thus, a quantitative analysis is impossible in this context, which is why we can only present visual results.
>
> * Discussion: The authors state "The requirement of point clouds is a mixed blessing, as it requires an additional step to work on image input", but the inference in the first experiment was run over positions sampled on a grid, which, to my understanding, would allow direct application to image data. Could the authors please clarify what they mean?
>   + Thank you for this question. This additional step refers to creating point clouds from images, which could be achieved directly through e.g. a learned approach, or by using image segmentations. We have slightly adapted the text.
>
> * Could the authors please clarify whether the spatial coordinates were normalized?
>   + We translated the coordinates such that the left ventricular apex is in the Cartesian origin. Other than that, there were no further steps for the normalization of the coordinates, which were given in millimeters in the original space.
>
> * Idea for future research: It would be very interesting to see whether, when trained on large amount of healthy and diseased shapes, the learned shape encodings are also discriminative of the underlying diseases. In addition, it would be very valuable to not only study the error under sparse data, but also partial shape data, e.g., providing only data on the septum or free wall
>   + Thank you for these suggestions. This looks like a very promising future research direction and we are grateful for bringing this to our attention.

---

> ### Author Response · Authors · 2024-03-16
> **Answer to the review 3/3**
>
> Point-by-point response to the additional questions:
>
> * Could the authors please explain whether and how this method could be extended to represent non-closed surfaces?
>   + Thank you for your question. The generalization to non-closed surfaces is beyond the scope of this paper. However, there exists some work addressing learned unsigned distance functions (e.g. “Neural Unsigned Distance Fields for Implicit Function Learning” by Julian Chibane, Aymen Mir, Gerard Pons-Moll, NeurIPS 2020).
>
> * In traditional SSMs one can retrieve the mean shape, do the authors see any way to retrieve something similar with the proposed approach?
>   + We assume a zero mean Gaussian prior, therefore the reconstruction of the zero-latent vector would provide the prior mean shape and would be easily computable. However, the computation of the mean shape in Cartesian space on the level of shapes is not as straightforward in our case, as it is for traditional SSMs.
>
> * Another benefit of SSMs are point correspondences, which can be easily used to transfer data between geometries. How could the proposed method be used for such tasks?
>   + To the best of our knowledge, there is no canonical approach for point correspondences for signed distance functions.
>
> * A shape encoding of 64 dimensions was chosen. Could the authors please comment on the impact of the latent dimension size onto the results?
>   + Thanks for the question. We experimentally discovered that a latent space dimension of 64 provides the best tradeoff between training accuracy and prediction quality. However, we found that the results are not very sensitive to changes in the latent space dimension. Our experiments reveal that the performance mainly depends on the choice of the Lipschitz regularization parameter (alpha) and the prior weighting (sigma).
>
> * Appendix, A2: Could the authors please clarify why two point clouds were generated? In addition, could the authors please comment on the impact of only sampling within a 30mm distance to the shape? What happens if one would sample further away? Could this be an issue for very large or very small geometries?
>   + Thank you for this comment. The two point clouds previously referred to endocardial and epicardial points respectively, but are later merged into a single point cloud. To avoid any further confusion, we only mention a single generated point cloud (left, right, epi- and endocardial).
>    The main aim of the proposed framework is the reconstruction of shapes by means of the signed distance function, which exhibits its highest accuracy in the neighborhood of the zero levelset. The Lipschitz regularity of the network bounds the change of the SDF in a rough sense, but the signed distance may still deviate from the true signed distance beyond the trained distance. Note however that in no experiment, we observed a second disconnected level set at another point in the discretized mesh grid, even though some portions in the bounding box exceed the distance of 30mm to at least one of the four shapes.

---

### Official Review · Reviewer_f255 · 2024-03-01

**Confidence:** 5
**Preliminary Rating:** 5
**Recommendation:** Oral

**Summary:**

Signed distance functions of ventricular surfaces are jointly represented using a neural field. The neural field is conditioned on both 3D coordinates as well as a latent vector, and can be trained using mesh models of the heart (points + SDF values). Additional Lipschitz regularization of the network during training ensures a smooth latent space, which allows interpolation between shapes in latent space.

**Strengths:**

- Paper is well-structured.
- Elaborate overview of previous work, putting the presented work in perspective.
- Thorough comparison to state-of-the-art methods. Once trained, the presented model can be used for shape completion with a small amount of datapoints and is more robust to noise than existing methods.
- Diverse experiments, demonstrating accuracy of the reconstructions, as well as interpolation through latent space and a comparison to existing methods.

**Weaknesses:**

The one thing I missed is the potential clinical impact of this work. The results look very promising and show potential in e.g. modeling time series and the deformation of the heart in cardiac gated CTs, as the shapes can be conditioned on space + time as well.
Additionally, more background information on the shapes included in the training data would be beneficial. Are these all healthy hearts, or are some of them diseased? What is the distribution of the included shapes and how does this influence the model's capability to represent shapes that were outside of the distribution? Can a model trained only on healthy hearts still represent diseased hearts?

**Detailed Comments:**

- I don't agree with the notation for the zero-levelset in Section 3.1, this implies that a signed distance function is invertible, which is generally not the case. I would write the zero level set as {$x \in \mathbb{R}^3 | f_S(x) = 0$}.
- The endocardium and epicardium seem to be nested in one another, did you observe violations of the anatomical requirement in some representations?
- What is shown in the right part of Figure 3? Are those multiple reconstructions or only the surface reconstruction resulting from the proposed method?
- It might be nice to visualize the different parts of the heart (e.g. in Figure 1) using different colors so it is clear which parts are separate for reader that are not very familiar with heart anatomy.
- If you have shapes for healthy and diseased hearts available, are they clustered in the latent space? Is this also influenced by the shapes that were present in the training data?
- The last sentence of the discussion is unclear. What are the concrete implications of optimal transport in latent space? I think there are much more straightforward applications of the proposed work, e.g. modeling of time series.

**Justification Of The Preliminary Rating:**

The manuscript is clearly structured, innovative and shows potential for a lot of new work regarding organ shape analysis and/or disease progression. The experiments prove the effectiveness of the method.

**Questions To Address In The Rebuttal:**

- Are there clusters in latent space based on healthy / diseased heart shapes?
- Does your method violate topological priors?

**Special Issue:**

Yes

---

> ### Author Response · Authors · 2024-03-16
> **Answer to the review**
>
> Thank you for your valuable feedback and the very positive assessment of our work. We are aware of the missing clinical validation of this work for the aforementioned examples. However, this is beyond the scope of this work, which, however, will be addressed in future research.
>
> A point-by-point response to the detailed comments:
>
> * The one thing I missed is the potential clinical impact of this work.
>    +  We are grateful for this suggestion. In general we think that the presented method has the potential to offer an easier method to reconstruct anatomical models, where imaging data is scarce, not present, or limited to specific imaging modalities ( e.g. electroanatomical mapping, echocardiography). We tried to highlight possible clinical applications in the last part of the discussion in the revised version.
>
> * Are these all healthy hearts, or are some of them diseased? What is the distribution of the included shapes and how does this influence the model's capability to represent shapes that were outside of the distribution? Can a model trained only on healthy hearts still represent diseased hearts?
>   + Thank you for this question. We have now clarified in the text that 20 geometries are from healthy subjects and 24 geometries are from patients with heart failure candidate to resynchronization therapy (see Appendix A). We have examined the patients without any prior assumption regarding the underlying probabilistic model. Already in this cohort, a large inter-patient variability can be observed. Out-of-distribution shapes with a completely different anatomy such as patients suffering from single ventricle defects will likely not be properly represented by the current model unless the training set is enlarged by these diseases.
>
> * I don't agree with the notation for the zero-levelset in Section 3.1, this implies that a signed distance function is invertible, which is generally not the case. I would write the zero level set as $\lbrace \mathbf{x} \in \mathbb{R}^3 \colon f_S(\mathbf{x})=0\rbrace$ .
>   + Thank you for your comment. We now denote the zero level set as suggested.
>
> * The endocardium and epicardium seem to be nested in one another, did you observe violations of the anatomical requirement in some representations?
>   +  During training and in all inferences we performed, no topological or anatomical violations were observed using our method.
>
> * What is shown in the right part of Figure 3? Are those multiple reconstructions or only the surface reconstruction resulting from the proposed method?
>   + In the right part of Figure 3, we show the left and right endocardial reconstruction of our method based on points on the left endocardial reconstruction. Because of the joint latent code for all surfaces, our method can provide a reasonable estimation of the other surfaces.
>
> * It might be nice to visualize the different parts of the heart (e.g. in Figure 1) using different colors so it is clear which parts are separate for reader that are not very familiar with heart anatomy.
>   + Thank you for this suggestion. In the revised version, we added colors to highlight the left and right ventricles in Figure 1.
>
> * If you have shapes for healthy and diseased hearts available, are they clustered in the latent space? Is this also influenced by the shapes that were present in the training data?
>   + Thank you for this comment. We performed a simple k-means on the final latent vectors, which did not result in clusters visually separating healthy and diseased patients.
>
> * The last sentence of the discussion is unclear. What are the concrete implications of optimal transport in latent space? I think there are much more straightforward applications of the proposed work, e.g. modeling of time series.
>   + Thank you for bringing this to our attention. We agree that optimal transport in latent space is not the most straight-forward extension of our work and have thus removed it from the last paragraph of the discussion and added a sentence on time dependent models.

---

### Author Response · Authors · 2024-03-16
**General Statement**

We wanted to take this opportunity to thank all the reviewers for their time and effort to evaluate our presented work. We did our best to address all raised points sufficiently. The revised version contains many changes that are highlighted in a different color in the document. Additionally, the document was restructured to more efficiently emphasize our novelty and contributions. Most notably, Figure 2, the comparison with SSM and the lower half of Figure 4 were moved to the Appendix. The results of the SSM in Figure 3 were now replaced with the original DeepSDF methodology results, more closely connected to other related works. Detailed comments and answers to the reviewers’ points are provided in the comment section of the reviews.

---

### Meta-Review · Area_Chair_jumP · 2024-04-04

**Recommendation:** Accept (Poster)
**Confidence:** 5

**Metareview:**

This paper raised quite some questions, and I appreciate that the authors and reviewers have actively engaged in the discussion during the rebuttal period. Reviewer 7Bgm raised concerns about the novelty of the work with respect to existing publications. Although the work does show similarities with the paper by Sander et al, I believe the authors have made their contributions - in particular the inclusion of Lipschitz constraints - sufficiently clear in the revised manuscript. This is an exciting work that will be of interest to the MIDL community and thus I recommend acceptance.

---

### Decision · Program_Chairs · 2024-04-06

Accept (Poster)